# Finite Element Analysis of Channel Screw and Conventional Plate Technique in Tile B2 Pelvic Fracture

**DOI:** 10.3390/jpm13030506

**Published:** 2023-03-10

**Authors:** Dejian Li, Hanru Ren, Xu Zhang, Rongguang Ao, Chengqing Yi, Baoqing Yu

**Affiliations:** 1Department of Orthopedics, Shanghai Pudong Hospital, Fudan University Pudong Medical Center, Shanghai 201300, China; 2Department of Trauma Orthopaedics, Shanghai East Hospital, Tongji University School of Medicine, Shanghai 201300, China; 3Department of Orthopedics, Shanghai Pudong New Area People’s Hospital, Shanghai 201202, China

**Keywords:** finite element analysis, channel screw, plate, pelvic fracture

## Abstract

Objective: This study aims to analyze the biomechanical characteristics of tile B2 pelvic fractures using finite element analysis when the superior ramus of the pubis was fixed by a plate or hollow screws in standing and sitting positions, respectively. Methods: A three-dimensional digital model of the tile B2 pelvic fracture was obtained by CT scanning the patient. The main ligament structure was then reconstructed based on the anatomical characteristics to create a finite element model of the tile B2 pelvic fracture. The posterior pelvic ring was fixed by sacroiliac joint screws, while the anterior ring injury of the superior ramus of the pubis was fixed by plates and hollow compression screws, respectively. The degrees of freedom of the bilateral acetabulum or two sides of the ischial tuberosity were constrained in the two models. A vertical load of 600 N was applied to the upper surface of the sacrum to measure the displacement and stress distribution of the pelvis in the standing and sitting positions. Results: The displacement distribution of both the healthy and the affected side of the pelvis was relatively uniform in both the plate group and the hollow screw group according to the finite element simulation results. The maximum displacement value in the sitting position was greater than the standing position, and the maximum displacement value of the hollow screw fixation was greater than that of the plate fixation. In the four groups of fixation models, the maximum displacement value of the pelvis in the hollow screw sitting position group was 1616.80 × 10^−3^ mm, which was greater than that of the other three groups, and in this group the total displacement value of the hollow screw in the anterior ring was 556.31 × 10^−3^ mm. The stress distribution of the pelvis in the various models was similar in the four groups of models, in which the maximum stress of the pelvis in the hollow screw sitting position group was the largest, which was 201.33 MPa, while the maximum stress in the standing position was 149.85 MPa greater than that in the sitting position of the hollow screw fixation. Conclusion: The anterior ring of patients with Tile B2 pelvic fractures fixed with hollow screws or plates in both standing and sitting positions can achieve satisfactory biomechanical results with significant safety margins for plates and screws.

## 1. Introduction

Pelvic fracture is one of the most common types of high-energy injury in clinical practice, of which Tile B2 pelvic fracture is a very common type of injury, consisting mainly of injuries to the anterior pelvic ring and the posterior pelvic ring of the same side [1,2]. Treatment is usually surgical as long as the patient can tolerate it, but the options for surgical treatment remain relatively large [3,4].

At present, sacroiliac joint screws have been widely accepted for posterior ring fixation of Tile B2 pelvic fracture [5]. However, the stability of the anterior pelvic ring is practically as important as the stability of the posterior pelvic ring, and the anterior pelvic ring is an important mechanical element for reinforcing the weight-bearing arch of the pelvis [6,7]. Currently, the fixation of the anterior pelvic ring is still unclear, and the placement of a plate through a modified ilioinguinal approach is one of the more commonly used internal fixation methods. Xiaowei Yu et al. concluded that the minimally invasive technique of dissecting the plate for the treatment of upper and lower pubis fractures has satisfactory results [8]. However, plate fixation of pubis fracture is not a good solution to all clinical problems. For example, in the treatment of distal pubis fracture, it is necessary to fix plate screws through the pubis symphysis, which will cause some complications for patients in the future. In recent years, attempts have been made to fix fractures of the superior ramus of the pubis using minimally invasive hollow compression screws, and this has also achieved excellent clinical results [9]. Although there have been some studies comparing anterior pelvic ring fixation modalities, there is still a lack of the relevant types of finite element analysis studies to elaborate on this issue.

The purpose of this study is to simulate the Tile B2 pelvic fracture with the help of finite element analysis and to compare the mechanical analysis of upper pubic bone fixation using plate screws and hollow compression screws in standing and sitting positions, respectively.

## 2. Materials and Methods

All methods in this study were performed in accordance with relevant guidelines and regulations. Approval was obtained from the Ethics Committee of Fudan University Affiliated Pudong Medical Center.

### 2.1. Study Equipment and Design Principles

Female patients with Tile B2 fractures were scanned using a CT device (Philips Brilliance, Philips Healthcare, the Netherlands; slice thickness, 0.3 mm), and the scanned data were stored in Digital Imaging and Communication in Medicine (DICOM) format. The 2D data in DICOM format were then imported into MIMICS 19.0 (Materialise, Belgium) for reconstruction to generate a Tile B2 fracture model. In addition, we set the material properties of the pelvic components and internal fixation materials and considered the materials to be homogeneous and isotropic, where the plates, screws, pedicle screws, and long screws were made of titanium; the material property parameters are shown in Table 1. The linear units were then created for simulation on the selected corresponding areas on the model surface according to the anatomical starting and ending positions of the pelvic ligaments, and the main ligament structures and anatomical properties were reconstructed by adding six main ligaments (sacrospinous ligament, sacrotuberous ligament, interosseous ligaments, sacroiliac anterior ligaments, sacroiliac dorsal ligaments, and arcuate pubic ligaments) whose material properties were set based on previous studies, as shown in Table 2 [10,11]. Based on this, a finite element analysis model of the Tile B2 pelvic fracture was developed.

### 2.2. Simulation of Two Types of Surgical Fixation Models

The hollow compression screw model, reconstructed plate, and screw models were further created in CATIA software, where the hollow compression screw was a short screw of 6.5 mm in diameter and the reconstructed plate screw was a reconstructed titanium plate and screw of 3.5 mm. All models were assembled in ABAQUS 6.11 (DASSAULT Inc., Vélizy-Villacoublay, FRANCE) software, and the overlap of the internal fixation within the bone was removed. Both surgical fixation models were fixed posteriorly by sacroiliac joint screws, which entered from the auricular surface margin of the sacroiliac joint to the medial cortical penetration of the superior margin of the lth sacral foramen, according to the bone channel principle. The fixation of the anterior pelvic ring included plate fixation and hollow compression screw fixation, in which the plate fixation was performed by placing one end of the reconstructed plate above the superior pubic branch and the other end fixed to the medial aspect of the anterior superior iliac spine, with three screws placed at each end of the titanium plate for fixation Figure 1A. When the hollow compression screw fixation was performed, the hollow compression screws were placed from the anterior aspect of the superior pubic branch obliquely outward and upward, and then passed from the posterior aspect of the anterior inferior iliac spine through the fracture line and superior acetabular rim. The fixation pattern is shown in (Figure 1B).

### 2.3. Loads and Boundary Conditions

In this study, the iliac bone and the contact relationship between the sacrum and sacroiliac joint were set as binding constraints. The contact relationship between the hollow compression screw and the bone was set as a binding constraint. The contact relationship between the screw and the bone was set as a binding constraint. The contact relationship between the steel plate and the bone was set as sliding friction. The contact relationship between the screw head and the steel plate was set as sliding friction. In the simulated standing position, the model constrained six degrees of freedom of the two-sided acetabulum and applied a vertical downward load of 600 N to the upper surface of the sacrum to simulate the upper body weight (Figure 2A). In the simulated standing and sitting position, the model constrained six degrees of freedom of two-sided ischial tuberosity and applied a vertical downward load of 600 N to the upper surface of the sacrum (Figure 2B) [12].

## 3. Results

### 3.1. Displacement Analysis of the Fracture Fixation Model

The models were divided into the anterior ring plate standing group, anterior ring plate sitting group, anterior ring hollow compression screw standing group, and anterior ring hollow compression screw sitting group. First, the maximum total displacement of the various fixation models occurred at the most proximal iliac ridge of the ilium and the proximal sacrum. In addition, the Tile B2 fracture model had a relatively uniform displacement distribution on both the sound and injured sides of the pelvis in both the plate group and the hollow compression screw group. In contrast, the overall displacement cloud showed that the maximum displacement value in the sitting position was greater than that in the standing position under the same anterior ring fixation method, while the maximum displacement value in the anterior ring hollow compression screw fixation was greater than that in the anterior ring plate fixation, with maximum displacement values in the following order: anterior ring hollow compression screw sitting group (1616.80 × 10^−3^ mm) > anterior ring hollow compression screw standing group (695.37 × 10^−3^ mm) > anterior ring plate sitting group (344.46 × 10^−3^ mm) > anterior ring plate standing group (118.53 × 10^−3^ mm) (Figure 3, Table 3).

In the model of the anterior ring plate group, the maximum total displacement of the anterior ring plate occurred at the proximal end of the plate, which was fixed to the iliac wing. The maximum total displacement of the anterior ring plate in the standing position was (22.66 × 10^−3^ mm), while the maximum total displacement of the anterior ring plate in the sitting position was (172.35 × 10^−3^ mm). In the model of the anterior ring hollow compression screw group, the maximum total displacement of the anterior ring hollow compression screw occurred at the nearest end of the hollow compression screw. The maximum total displacement of the anterior ring hollow compression screw in the standing position was (53.77 × 10^−3^ mm), while the maximum total displacement of the anterior ring hollow compression screw in the sitting position was (556.31 × 10^−3^ mm). This showed the order of the maximum displacement values of the anterior ring implants as follows: anterior ring hollow compression screw in the sitting position > anterior ring plate in the sitting position > anterior ring hollow compression screw in the standing position > anterior ring plate in the standing position (Figure 4, Table 3).

### 3.2. Stress Analysis of the Fracture Fixation Model

The stress distribution in the Tile B2 pelvis fracture model was basically similar in different modes of fixation, but there were differences in the stress distribution between different fixation modes and postures. The overall maximum stresses of the anterior ring plate standing group and the anterior ring hollow compression screw standing group appeared above the median sacral crest (Figure 5A,B), with values of 57.72 MPa and 74.03 MPa, respectively. However, the overall maximum stresses in the anterior ring plate sitting group and the anterior ring hollow compression screw sitting group appeared near the median sacral crest, the sciatic spine, and the lesser sciatic notch (Figure 5C,D), with values of 56.49 MPa and 201.33 MPa. In addition, the stress distribution of patients fixed with a plate is relatively uniform (Figure 6), with maximum stresses of 21.11 MPa and 32.93 MPa, whereas the stresses of hollow compression screws in patients were concentrated at the distal threads of the hollow compression screw (Figure 7, Table 4), with maximum stresses of 137.81 MPa and 149.85 MPa, respectively, which were much higher than the maximum stresses in plate groups.

## 4. Discussion

Pelvic fracture is rather uncommon in clinical practice, but due to its specific location, it often causes damage to intra-abdominal organs and vital blood vessels, which ultimately leads to high disability and mortality rates [13,14]. Tile B2 pelvic fracture is the most frequent type of pelvic fracture and is usually a joint injury of the anterior and posterior rings of the pelvis caused by lateral violence. To restore the integrity and stability of pelvic rings, of which the posterior ring accounts for 60% and the anterior ring 40%, surgical is used to treat pelvic fractures [15]. Therefore, the majority of scholars currently consider the stability of the posterior pelvic ring to be more important, thus neglecting the treatment of anterior ring fractures. In contrast, anterior pelvic ring fractures are more common than posterior fractures in our clinical work. If the internal fixation of the anterior pelvic ring fracture fails, the pelvic rings will lose their balance, which can cause damage to important blood vessels and even death by hemorrhage. In addition, clinicians are paying more attention to the fixation of anterior pelvic ring fractures since it can effectively strengthen the stability and resistance of the pelvis.

The most popular surgical approach for fixing the anterior pelvic ring is the internal fixation with an incisional repositioning plate [16]. This method requires making a transverse incision of approximately 4 cm above the pubic symphysis and 2 cm behind the anterior superior iliac spine, as well as implanting a pre-curved reconstruction plate through a subcutaneous tunnel to fix the superior pubic branch. This procedure fixes stability, but it comes with a lengthy surgical incision, relatively high intraoperative bleeding, and the possible risk of important neurovascular collateral damage. In contrast, more and more patients can now choose minimally invasive surgical methods for anterior ring fixation thanks to the advent of digital smart orthopedics and the minimally invasive concept [8,9,17]. A common surgical approach for the treatment of Tile B2 pelvic fractures involves the minimally invasive pedicle screw fixation of the sacroiliac joint in conjunction with hollow compression screw fixation of the pubic branch fracture, requiring only a 1 cm surgical incision, minimal bleeding, a shorter operative time, and smaller surgical scars. However, hollow compression screws need to be inserted through the suprapubic branch and the anterior column of the acetabulum, which is usually irregularly curved and has a small diameter in the intramedullary region and the anterior column, making the surgery extremely difficult with an extremely long learning curve. However, this issue has gradually been resolved since the introduction of the computer-assisted navigation system. With the aid of the O-arm machine’s navigation, we can reposition and fix the jugular intraoperatively, determine the screw position in real-time, lessen the secondary injury of surgery and medical radiation from repeated fluoroscopy, and shorten the operation time [18,19].

In our study, a Tile B2 pelvic fracture model (unilateral sacroiliac joint injury and superior and inferior pubic branch fracture) was established using the finite element method. The posterior sacroiliac joint injury was fixed with sacroiliac screws, while the anterior pubic branch fracture was fixed with hollow compression screws and plates, respectively. All fixation models were viewed as being reduced anatomically. Furthermore, based on previous studies, any displacement greater than 10 mm in the study was regarded as a poor prognostic indicator, while a displacement of less than 5 mm was regarded as acceptable and a displacement of 0.1 to 1 mm was regarded as a displacement that could shorten the healing time with a small range of motion. Naturally, the smaller the maximum displacement, the greater the postoperative stability of the fracture. As a result, both anterior ring hollow compression screw fixation and plate fixation provide adequate mechanical stability, and the overall stability is higher with plate fixation than with hollow compression screw fixation.

We also noticed that the displacement value of the pelvis in the hollow compression screw sitting group was 1616.80 × 10^−3^ mm, which was much larger than the maximum total displacement values of the other three groups, indicating that the maximum total displacement value of the pelvis was larger when the patient was sitting after hollow screw fixation. In this regard, we believe that the main reason is related to mechanical conduction, as the pressure on the pelvis from the spine in the standing position is transmitted directly to the bilateral acetabulum through the posterior pelvic ring, and only a small part of the force is dispersed to the anterior pelvic ring, the suprapubic branch, and pubic symphysis. In the seated position, the pressure from the spine is transmitted to the sciatic tuberosity, where more force is distributed to the superior pubic ramus and anterior pelvic ring. Hollow nail fixation is less stable than plate fixation, and pelvic maximum total displacement increases significantly when more stress is encountered.

The pelvis stress analysis revealed that the use of the anterior ring screws was safe and reliable from a stress perspective and that the stress results of the plates and screws in both standing and sitting positions were significantly lower than the yield strength of 860 MPa; still, the pelvis stress result has a large safety margin of comparison with the pelvis biomechanical results of Maslov et al. [20,21]. However, the stress distribution of the iliac bilaterally was more consistent, which indicated that the stress in the posterior sacral area on the affected side was significantly higher than that in the posterior sacral area on the sound side, and this was significantly related to the posterior sacroiliac joint injury. Undoubtedly, plate fixation had better stability compared to anterior ring hollow compression screw fixation and less maximum stress on both the pelvis and the internal fixator. However, we also observed that the stress of 149 MPa in the hollow compression screw during fixation in the standing position was greater than 74 MPa in the full pelvis, implying that hollow compression screws were better able to withstand the body stress in the standing position. This may be because hollow compression screws were fixed intramedullary, and the plate was fixed eccentrically. In addition, there are no biomechanical studies on the type of fractures, although our findings are consistent with the maximum stress values currently used for postoperative pelvic fractures, which vary from 15–250 MPa.

We all know that Tile B2 pelvic fracture is a rotationally unstable but vertically stable pelvic fracture. Lateral stress may lead to internal fixation failure following fracture surgery; hence, it is best to avoid lateral lying after surgery. In addition, Lan Li et al. analyzed the mechanical environment at 30°, 60°, and 90° in semi-recumbency after pelvic fracture surgery, and the results showed that the fracture displacement of patients seated at 90° was small and that there was no substantial mechanical concentration [12]. Therefore, we expect that patients with Tile B2 pelvic fracture can perform standing and sitting activities until the fracture heals within 3 months of surgery. For this purpose, we undertook a biomechanical analysis of patients in both standing and sitting positions, and the results also showed that both standing and sitting patients have greater safety margins of the pelvis and internal fixation to satisfy daily needs.

Based on our findings, we draw the conclusion that patients with Tile B2 pelvic fracture can achieve satisfactory biomechanical outcomes in both standing and sitting positions when the posterior ring is fixed with sacroiliac screws and the anterior ring is fixed with hollow compression screws or plates, with a greater margin of safety for plates and screws. Of course, plate fixation yields more stable biomechanical results than screw fixation does, but postoperative stability is by no means the only criterion for successful fracture surgery. Other factors to take into account include the size of the surgical trauma, the duration of the surgery, the availability of appropriate surgical instruments, and the surgical habits of different surgeons. The surgeon needs to choose the appropriate fixation method according to the patient’s condition.

## 5. Conclusions

The limitations of this study are as follows: (1) The effect of muscle strength and synovial condition on pelvic stability is still lacking in our study model, mainly because muscle strength is difficult to homogenize as it varies from patient to patient. (2) Some scholars believe that no additional fixation is needed for the anterior ring and only the posterior ring needs to be fixed, while the group with the unfixed anterior ring was added to our study for comparison and to observe the stability of pelvic biomechanics to determine if an additional fixation of the anterior ring is needed. (3) The Tile B2 pelvic fracture is a rotationally unstable fracture, but patients also need to rotate and move after surgery, and we did not analyze the biomechanical environment of patients in the lateral position.

## Figures and Tables

**Figure 1 jpm-13-00506-f001:**
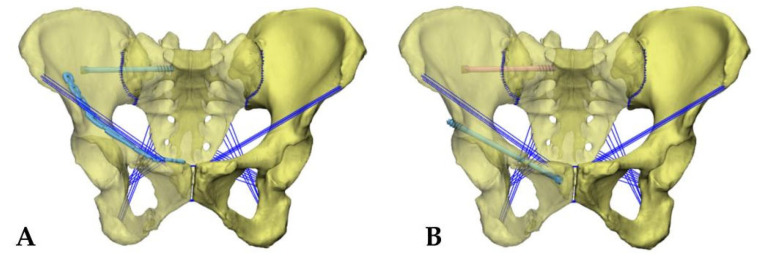
(**A**) FE model Tile b2 pelvic fractures fixed with sacroiliac joint screws and anterior pelvic ring plate. (**B**) FE model Tile b2 pelvic fractures fixed with sacroiliac joint screws and anterior pelvic ring hollow screw.

**Figure 2 jpm-13-00506-f002:**
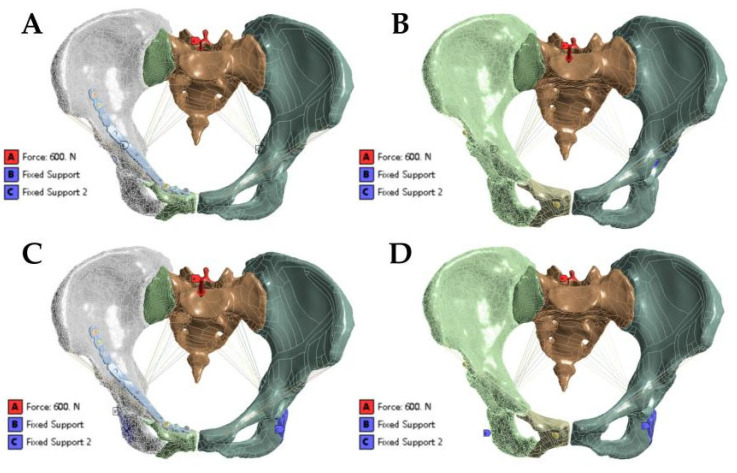
(**A**,**B**) Loads and boundary conditions of anterior ring plate group and anterior ring hollow screw group in standing position. (**C**,**D**) Loads and boundary conditions of anterior ring plate group and anterior ring hollow screw group in sitting position.

**Figure 3 jpm-13-00506-f003:**
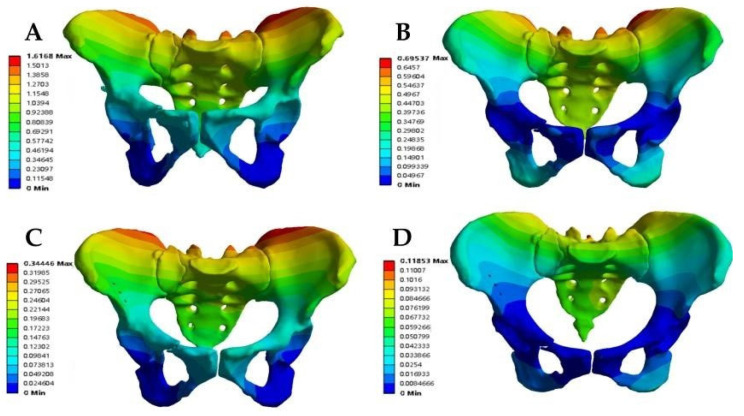
The maximum total displacement in Tile b2 pelvic fracture under different fixation modes ((**A**) anterior ring hollow compression screw sitting group; (**B**) anterior ring hollow compression screw standing group; (**C**) anterior ring plate sitting group; (**D**) anterior ring plate standing group).

**Figure 4 jpm-13-00506-f004:**
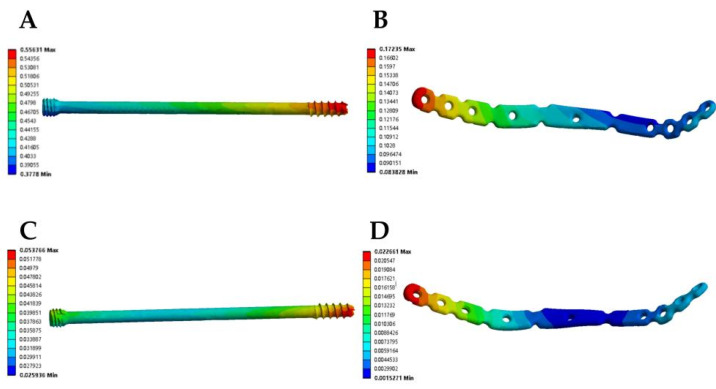
The maximum total displacement in Tile b2 pelvic fracture under different fixation modes ((**A**) anterior ring hollow compression screw sitting group; (**B**) anterior ring hollow compression screw standing group; (**C**) anterior ring plate sitting group; (**D**) anterior ring plate standing group).

**Figure 5 jpm-13-00506-f005:**
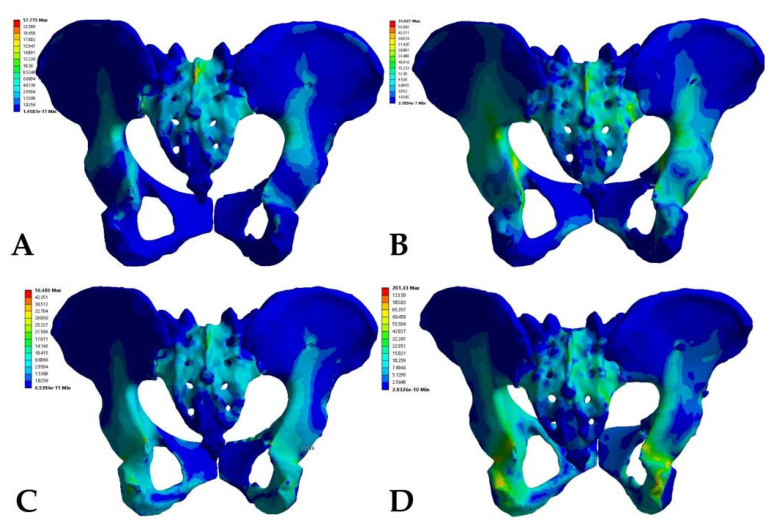
Von Mises stress distribution of pelvic models fixed with different implants in standing position. (**A**) Anterior ring plate. (**B**) Anterior ring hollow screw. Von Mises stress distribution of pelvic models fixed with different implants in sitting position. (**C**) Anterior ring plate. (**D**) Anterior ring hollow screw.

**Figure 6 jpm-13-00506-f006:**
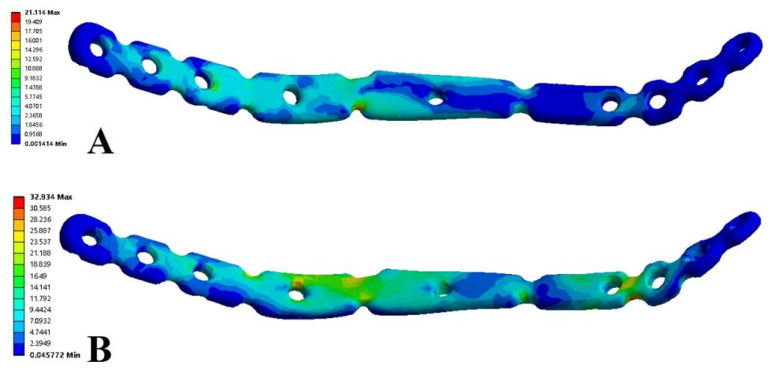
Von Mises stress distribution of implant during fixed with anterior ring plate. (**A**) Standing position. (**B**) Sitting position.

**Figure 7 jpm-13-00506-f007:**
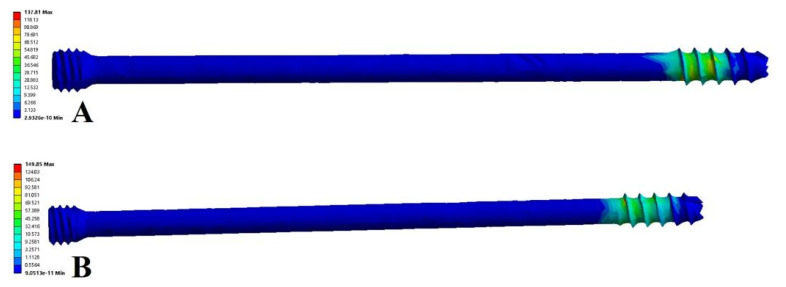
Von Mises stress distribution of implant fixed with anterior hollow screw. (**A**) Standing position. (**B**) Sitting position.

**Table 1 jpm-13-00506-t001:** Material properties of each part of the skeleton model.

Area and Material	Young’s Modulus (MPa)	Poisson’s Ratio
Sacrum cortical	17,000	0.3
Sacrum cancellous	150	0.2
Ilium cortical	17,000	0.3
Ilium cancellous	150	0.2

**Table 2 jpm-13-00506-t002:** Main ligament parameters of the pelvis.

Ligament	Stiffness Coefficient (N/mm)	Ligament
Sacroiliac	5000	Sacroiliac
Sacrospinous	1500	Sacrospinous
Sacrotuberous	1500	Sacrotuberous
Inguinal	250	Inguinal
Superior pubic	500	Superior pubic
Arcuate pubic	500	Arcuate pubic

**Table 3 jpm-13-00506-t003:** The maximum displacement (MD) of different finite element models after loading the pelvis.

Model	Pelvis MD (mm)	Anterior Ring Implants MD (mm)
anterior ring hollow compression screw (sitting)	1616.80 × 10^−3^	556.31 × 10^−3^
anterior ring anterior ring plate (sitting)	344.46 × 10^−3^	172.35 × 10^−3^
anterior ring hollow compression screw (standing)	695.37 × 10^−3^	53.77 × 10^−3^
anterior ring anterior ring plate (standing)	118.53 × 10^−3^	22.66 × 10^−3^

**Table 4 jpm-13-00506-t004:** The maximum stresses (MS) of different finite element model after loading the pelvis.

Model	Pelvis MS (MPa)	Anterior Ring Implants MS (MPa)
anterior ring hollow compression screw (sitting)	201.33	149.85
anterior ring anterior ring plate (sitting)	56.49	32.93
anterior ring hollow compression screw (standing)	74.03	137.81
anterior ring anterior ring plate (standing)	57.72	21.11

## Data Availability

The reader can obtain research data from the corresponding author.

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
