# Peer review of "Finite Element Analysis of Channel Screw and Conventional Plate Technique in Tile B2 Pelvic Fracture"

_jpm, 2023, doi:10.3390/jpm13030506_

Round 1

Reviewer 1 Report

The paper “Finite Element Analysis of Channel Screw and Conventional 3 Plate Technique in Tile B2 Pelvic Fracture” is an original research on treatment of pelvic fractures using finite element analysis with plates and screws and standing and sitting positions. It explains clearly about the research methodology and the results. However, there are certain queries that need to be addressed.

1.       What is the rationale for selecting only female patients. Please explain.

2.       Please provide tabular form of the data for stress analysis for different attributes.

3.       “The displacement value of the pelvis in the hollow screw sitting position group was 1616.80×10-3 mm, which was greater than that of the other three groups, and in this group the total displacement value of the hollow screw in the anterior ring was 556.31×10-3 mm” – what is the reason for this behavior. Please elaborate in the discussion.

Author Response

Point 1: 1. What is the rationale for selecting only female patients. Please explain.

Response 1: Our study required the medical records of only one patient with a typical pelvic TILE 2 fracture, and the female patient was selected for this analysis because this patient is more typical of the many pelvic TILE 2 fracture models. In addition, the female pelvis is slightly different from the male pelvis in terms of biomechanical studies, and some studies suggest that the maximum stress in women is less than that in men. Therefore, we chose the female pelvis as the typical pelvic TILE 2 fracture model.

Point 2: Please provide tabular form of the data for stress analysis for different attributes.

Response 2: Thank you for your comments. We have added the table 3 and table 4 in our manuscript.

Point 3: “The displacement value of the pelvis in the hollow screw sitting position group was 1616.80×10-3 mm, which was greater than that of the other three groups, and in this group the total displacement value of the hollow screw in the anterior ring was 556.31×10-3 mm” – what is the reason for this behavior. Please elaborate in the discussion.

Response 3: Thank you for your comments. We have added it in our manuscript.

Reviewer 2 Report

This paper presents finite element analysis to examine the biomechanical properties of pelvic fractures. By CT scanning the patient, a three-dimensional digital model of the pelvic fracture was created. According to the outcomes of the finite element simulation, the displacement distribution of the healthy and damaged sides of the pelvis was fairly uniform in both the plate group and the hollow screw group. While the maximum displacement value of the hollow screw fixation was more than that of the plate fixation, the maximum displacement value in the sitting position was greater than that in the standing position.

Comments to authors:

1. Fig 1 and Fig 2 combine into one figure.

2. Fig 3 and Fig 4 combine into one figure.

3. Fig 6 - larger

4. Fig 7 and Fig 8 combine into one figure.

5. In Discussion section there is no comparison with data published in the recent Maslov et al paper

https://vestnik.pstu.ru/biomech/archives/?id=&folder_id=11116

Author Response

Point 1: 1. . Fig 1 and Fig 2 combine into one figure.

Response 1: Thank you for your comments. We have combine Fig 1 and Fig 2 into one figure.

Point 2: Fig 3 and Fig 4 combine into one figure.

Response 2: Thank you for your advice. We have combine Fig 3 and Fig 4 into one figure.

Point 3: Fig 6 - larger

Response 3: Thank you. We have adjusted the size of Fig 6.

Point 4. Fig 7 and Fig 8 combine into one figure.

Response 4: Thank you for your comments. We have combine Fig 7and Fig 8 into one figure.

Point 5. In Discussion section there is no comparison with data published in the recent Maslov et al paper

https://vestnik.pstu.ru/biomech/archives/?id=&folder_id=11116

Response 5: Thank you very much. According to your suggestions, we have modified the article and added corresponding references.
